# Hyperoxia Alters Ultrastructure and Induces Apoptosis in Leukemia Cell Lines

**DOI:** 10.3390/biom10020282

**Published:** 2020-02-12

**Authors:** David De Bels, Frauke Tillmans, Francis Corazza, Mariano Bizzarri, Peter Germonpre, Peter Radermacher, Keziban Günce Orman, Costantino Balestra

**Affiliations:** 1Environmental and Occupational Physiology Laboratory, Haute Ecole Bruxelles-Brabant, 1180 Brussels, Belgium; ftillmans@dan.org (F.T.); pgermonpre@gmail.com (P.G.);; 2Intensive Care Department, Brugmann University Hospital, 1020 Brussels, Belgium; 3Translational Research Laboratory, Université Libre de Bruxelles, 1050 Brussels, Belgium; francis.corazza@chu-brugmann.be; 4Clinic of Anesthesiology, Section Anaesthesiological Pathophysiology and Process Development, University of Ulm, 89081 Ulm, Germany; peter.radermacher@uni-ulm.de; 5Immunology Laboratory, Brugmann University Hospital, 1020 Brussels, Belgium; 6Experimental Medicine Llaboratory, La Sapienza University of Rome, 00185 Rome, Italy; mariano.bizzarri@uniroma1.it; 7Hyperbaric Centre, Queen Astrid Military Hospital, 1120 Brussels, Belgium; 8DAN Europe research (1160 Auderghem, Brussels, Belgium; 64026 Montepagano, Roseto, Italy); 9Department of Computer Engineering, Galatasaray University; Ortakoy, 34349 Istanbul, Turkey; gunceorman@gmail.com; 10Motor Sciences Department, Physical Activity Tteaching Unit, Université Libre de Bruxelles (ULB), 1050 Brussels, Belgium

**Keywords:** hyperoxia, leukemia, normobaric oxygen paradox, fractals, caspase, apoptosis

## Abstract

Oxygenation conditions are crucial for growth and tumor progression. Recent data suggests a decrease in cancer cell proliferation occurring after exposure to normobaric hyperoxia. Those changes are associated with fractal dimension. The purpose of this research was to study the impact of hyperoxia on apoptosis and morphology of leukemia cell lines. Two hematopoietic lymphoid cancer cell lines (a T-lymphoblastoid line, JURKAT and a B lymphoid line, CCRF-SB) were tested under conditions of normobaric hyperoxia (FiO_2_ > 60%, ± 18h) and compared to a standard group (FiO_2_ = 21%). We tested for apoptosis using a caspase-3 assay. Cell morphology was evaluated by cytospin, microphotography after coloration, and analysis by a fractal dimension calculation software. Our results showed that exposure of cell cultures to transient normobaric hyperoxia induced apoptosis (elevated caspase-3) as well as significant and precocious modifications in cell complexity, as highlighted by increased fractal dimensions in both cell lines. These features are associated with changes in structure (pycnotic nucleus and apoptosis) recorded by microscopic analysis. Such morphological alterations could be due to several molecular mechanisms and rearrangements in the cancer cell, leading to cell cycle inhibition and apoptosis as shown by caspase-3 activity. T cells seem less resistant to hyperoxia than B cells.

## 1. Introduction

Cancer cells usually display a hypermetabolic fingerprint that can be sustained even in the presence of low/normal oxygen supply by adopting a glycolytic nonoxidizing metabolic phenotype [1]. Reduced dependence on oxygen is not only necessary for activating the so-called Warburg effect, but also has a pivotal role in decreasing the production of reactive oxidant species (ROS), which can be potentially harmful to cancer cell viability. While hypoxia has been extensively studied in cancer, only recent literature has investigated the possible role of hyperoxia in tumor necrosis [2]. Hyperoxia can in fact be deleterious to cancer cells in different ways. Some authors [3,4] have demonstrated that apoptosis can result from hyperoxia on solid tumors, while others have shown cell cycle blocking in carcinoma cells [5] or anticancer immune-surveillance due to T cell and NK cell stimulation [6]. Hyperoxia increases the level of ROS inside the cell and has a direct effect on both the cell cycle and cell viability. ROS can induce heavy damage to DNA, thus increasing cell cycle length because of multiple repair needs [7].

Hypoxia-inducible factors (HIF-1 or perhaps more so HIF-2) [8,9,10,11,12] are implicated in the mechanisms of cell apoptosis, both inhibiting [13] and fostering [14,15,16,17,18,19] them. It is noteworthy that HIF has been shown to be under-expressed during stable hyperoxia and over-expressed just after return to normoxia in noncancer (HUVEC) cells, confirming the fact that transient hyperoxia is perceived as a hypoxic stimulus [20]. This is an unexpected finding as HIF has been extensively studied only under hypoxic conditions [21].

It is tempting to speculate that inducing changes in HIF expression by hyperoxia could lead to biological changes in cells with a high metabolic demand.

It has been shown that normobaric oxygen significantly increases endogenous erythropoietin (EPO) production 10, 12, 24, and 36 h after oxygen administration [22], hence fostering hemoglobin increase in healthy volunteers [23]. Latency time seems consistent with the time necessary to transcribe, translate, produce, and secrete erythropoietin. Indeed, erythropoietin (EPO) and vascular endothelial growth factor (VEGF) production are under the control of the HIF transcription factor [24,25]. Combined with chemotherapy, normobaric oxygen has recently been shown to reduce the tumor load and number in mice with lung cancer [26]. The mechanism by which hyperoxia could interfere with cell survival lies deep in the fundamental cellular mechanisms of adaptation to hypoxia as proposed in Figure 1 [27]. However, being harmful inside cells is not the only way by which hyperoxia could be deleterious to cancer cells. Indeed, another way that hyperoxia could have antitumor activity is by inducing numerous modifications in the adenosine pathway, activating anticancer effects of T cells and NK cells [6,28]. Hyperoxia is also known to mobilize stem progenitor cells and change cytokine expression [29].

The purpose of the present study was to investigate the role of transient hyperoxia on the outcome of nonsolid tumor cells such as leukemia line cells. This was performed using a biological marker (caspase-3), a morphological microscopic analysis, and a nonlinear fractal dimension calculation.

## 2. Material and Methods

### 2.1. Cell Lines

CCRF-SB and Jurkat leukemia cell lines (Sigma-Aldrich, St Louis, MO, USA) were maintained in RPMI 1640 containing 10% FCS, 100µg/mL Penicillin, and 100µg/mL Streptomycin in a fully humidified incubator at a concentration of 0.3 × 10^6^ cells/mL under standard conditions of 21% O_2_, 5% CO_2_, and 37 °C. 

For incubation in an increased oxygen environment, an oxy-concentrator was used (Generator 6000, b-Cat, Tiel, The Netherlands). Cells were incubated in 60% O_2_, 5% CO_2_, and 37 °C for 18h and returned to normoxia for the following incubation. Cells were sampled from cultures at various times of incubation for apoptosis and cell cycle determination. In selected experiments, hyperoxia incubation was repeated. The medium was kept unchanged.

### 2.2. Protein Extraction and Western Blot

For the Western Blot analysis, cells were centrifuged at 1200 rpm for 5 min. The pelleted cells were washed twice in PBS and then lysed (Gibco Lysis Buffer, with the addition of Antiprotease/Antiphosphatase) for 10 min on ice.

After centrifugation at 13,000 rpm for 10 min, the supernatant (protein extract) was immediately frozen to −80 °C. For the assessment of the expression of Bcl-xL protein, protein concentrations were determined, and equal total protein aliquots of 20 µg/10 µL were separated by sodium dodecyl sulfate polyacrylamide gel electrophoresis and transferred by Western blotting. After blocking, the membranes were incubated with commercially available primary Bcl-xL antibody (Cell Signaling Technology, Danvers, MA, USA). The primary antibody was detected using horseradish peroxidase conjugated secondary antibodies (Cell Signaling Technology, Danvers, MA, USA) and the membranes were subjected to chemoluminescence using the SuperSignal West Femto Maximum Sensitivity Substrate (Thermo Scientific, Dreieich, DE, USA). Exposed films were scanned and intensity of immunoreactivity was measured using NIH ImageJ software (http://rsb.info.nih.gov/nih-image). Data are expressed as fold increase over control values.

### 2.3. Caspase-3

Caspases were analyzed by a Western blot analysis. For immunoblot analyses, 40 µg of protein lysates per sample were denatured in SDS-PAGE sample buffer (Tris-HCl 260 mM, pH 8.0, 40% (*v*/*v*) glycerol, 9.2% (*w*/*v*) SDS, 0.04% bromophenol blue and 2-mercaptoethanol as a reducing agent) and subjected to SDS-PAGE on 5% acrylamide/bisacrylamide gels. Separated proteins were transferred to nitrocellulose membrane (Hybond-P PVDF, Amersham Biosciences). Residual binding sites on the membrane were blocked by incubation in TBST (10 mM Tris, 100 mM NaCl, 0.1% Tween 20) with 5% (*w*/*v*). Membranes were then probed with a specific primary antibody, Cleaved Caspase-3 (Cell Signaling Technology, Danvers, MA, USA) (1:200). This was followed by a peroxidase-conjugated secondary antibody, HRP labeled mouse antirabbit Ig (Cell Signaling Technology, Danvers, MA, USA) (1:10000), and visualized with an ECL Plus detection system (Amersham Biosciences). The equivalent loading of proteins in each well was confirmed by Ponceau staining [30].

### 2.4. Pictures of Thin Layer Cell Preparation

Using a Cytospin 4 cytocentrifuge (Thermo Scientific, Waltham, MA, USA), 200 µL of cell culture exposed to normobaric hyperoxia or control culture from normoxic conditions were sampled and centrifuged for 5 min at 600 rpm on microscope slides, followed by a standard Giemsa coloration. The slides were then read under a light microscope with 50-fold magnification.

Cell lines cultured in RPMI 1640 supplemented with fetal calf serum were collected. Slides were prepared from cell suspension by cytocentrifugation and stained with May-Grünwald-Giemsa. Randomly chosen fields with non-overlapping cells were captured using a Leica Dialux EB 20 microscope (oil immersion objective ×100) equipped with a DP200 digital camera (DetaPix).

For each slide, classical morphometric parameters such as nuclear area appearance (condensed or uncondensed chromatin, nuclei cytoplasmic ratio, presence of nucleoli, and regularity of nuclei) and cell cytoplasm color and regularity of contour were assessed as well as the presence of apoptotic or mitotic pictures. A scoring system based on FAB criteria [31] was attributed to each morphologic item (from 0 for absence to 4 for very important). The biologist was blinded as to whether the cells had or had not been submitted to hyperoxia.

### 2.5. Fractals

For the purpose of the present investigation, determination of the fractal dimension of cells was performed using the Harfa 5.5 program (Faculty of Chemistry, Brno University of technology, Brno, Czech Republic) and by applying the box counting method after appropriate filtering and threshold application. The accepted final result was taken as the fractal dimension with the best fit to the slope described in the slope analysis (Figure 2).

A fractal (from the Latin ‘fractus’, ‘broken’) is an object with a noninteger dimension that looks exactly the same at every scale. However, the definition of *fractal* goes beyond self-similarity per se to exclude trivial self-similarity and include the idea of a detailed pattern repeating itself. Fractal patterns with various degrees of self-similarity have been studied in images, structures, and found in nature and technology [32].

Euclidean descriptions are not adequate for complex irregular-shaped objects that occur in nature. These “non-Euclidean” objects are better described by fractal geometry, which has the ability to quantify the irregularity and complexity of objects with a measurable value called the fractal dimension [33].

A geometrically intuitive notion of dimension is as an exponent that expresses the scaling of an object’s bulk with its size:bulk (N) = size^dimension^(1)

Here, bulk may correspond to a volume, a mass, or even a measure of information content and size is a linear distance (Figure 3).

For example, the area (bulk) of a plane figure scales quadratically with its side (size), and so it is two dimensional, meanwhile a volume is related to the cube of its side. By transforming such relationships through the use of logarithms, we obtain a general equation of the form
(2)$$\mathrm{Dimension}\;=\;\frac{\lim}{size\to 0}\;\frac{\log bulk}{\log size}$$
where size is generally expressed as a fraction of the entire bulk: 1/N = K. This ratio is generally known as homothetia, meaning the operation able to geometrically transform the space without changing its form, i.e., preserving the pattern in between its constitutive elements. Bulk can be divided in N fractions (similar to the entire bulk), and each of those fractions has a length equal to 1/N=K. Then, we obtain D_segment_ = $\frac{\log N}{\log N}$= 1, meanwhile for an area we have D_area_ = $\frac{\log N^{2}}{\log N}\;$= 2. For a fractal object, like the Koch snowflake (Figure 3) we have four segments similar to the entire bulk, each one equal to 1/3 of the entire length. Thus, the (fractal) dimension of that object can be calculated as D= $\frac{\log4}{\log3}$ = 1.262. Its fractal dimension (1.262) therefore exceeds its topological dimension ‘1’, providing a quantitative measure of the space-filling capacity of a pattern that tells how a fractal scales differently than the space it is embedded in [34]. Dimension is mathematically expressed by so-called “power laws”, since the Equation (1) shows that some quantity N can be expressed as some power of another quantity, *s:*N_(s)_ = *s*^-τ^

Taking the logarithm on both sides of the equation, we find a relationship indistinguishable from (2). By plotting log *N_(s)_* versus log *s* we obtain a straight line (the signature of the power law), being τ (a noninteger number) the slope of the straight line. The scale invariance can be seen from the fact that the straight line looks the same everywhere.

### 2.6. Statistics

Standard statistical analyses were performed, including mean, standard deviation, and ANOVA for repeated measures to test the between- and within-subject effect after Kolmogorov-Smirnov testing for normality. The Bonferroni or Dunnett’s tests were applied to the experimental and control values. Taking the initial value as 100%, percentual variations were calculated for each parameter, thereby allowing an appreciation of the magnitude of change rather than the absolute values.

Other tests between groups such as t-tests (with Welch correction) or non-parametric analysis were done when appropriate (Mann-Whitney, Wilcoxon). All statistical analyses were performed with the GraphPad Prism version 8.31 for Windows (GraphPad Software, La Jolla, CA, USA).

## 3. Results

Evident morphological modification of cell shape in JURKAT cells after 18 h of hyperoxic stress can be seen in Figure 4.

Caspase-3 activity increased after hyperoxia and was even further elevated after return to normoxia when compared to data obtained in leukemia cell lines not submitted to transient hyperoxia (Figure 5). Bcl-xl activity significantly increased 6 and 12h after hyperoxia in CCRF-CB line cells but remained statistically unchanged in JUKAT cell lines (Figure 6).

However, morphological changes are generally not always so obvious (see Figure 2); it is precisely in these situations that the usefulness of a fractal approach emerges in assessing how living structures have changed. 

Fractal dimensions significantly increased in CCRF-SB and JURKAT cell lines after hyperoxia. Figure 7 illustrates the fractal dimension variation in cell lines (JURKAT, Figure 7A and CCRF-SB, Figure 7B) up to 48 h after an 18-h hyperoxic period, compared to normoxia. A gradual increase in fractal dimensions of JURKAT (Figure 7A) and CCRF-SB cells (Figure 7B) can be seen along the course of the experiment from 0 to 48 h in culture medium following 18 h under hyperoxic conditions. There was no statistical difference between the different groups at 0 and 2 h. Both cells in the standard and hyperoxia group increased their fractal dimensions, but cells submitted to hyperoxia had significantly higher fractal dimensions at 4, 6, and 48 h following 18 h of hyperoxia. 

Morphological changes revealed a significantly higher number of pycnotic nuclei (Figure 8A; *p* = 0.0058) in the cells submitted to hyperoxia. Moreover, the visual analysis confirmed the significantly higher number of apoptotic cells in the hyperoxia group (Figure 8B; *p* = 0.036) as previously suggested by the caspase results.

## 4. Discussion

### 4.1. Apoptosis Hypothesis

Cells in the hyperoxia group expressed a twelve-fold increase in caspase-3 values as compared to the control group. This seems to indicate that cancer cells submitted to hyperoxia are committed to apoptosis. Early stages of apoptosis, as seen here, have been associated with increased values of both membrane and chromatin fractal dimension [35]. It is likely that such changes in cellular morphology may involve a reorganization of the cytoskeleton, and could follow the induction of apoptosis (i.e., mitochondrial fission). Hyperoxia leads to mitochondrial damage [36] and, as outlined by Chalut and coworkers [35], mitochondrial structural changes are at least partly responsible for the early membrane fractal changes observed during apoptosis, whereas nuclear substructure modifications are likely responsible for later changes in the fractal dimension of subcellular systems. Given that hyperoxia is thought to induce apoptosis through mitochondrial damage, it is tempting to speculate that such morphological modifications may reflect the complex metabolic deregulation underlying the mitochondrial involvement taking place at the very beginning of the apoptotic pathway.

### 4.2. Fractal Analysis

Several articles have documented that fractal values increase significantly at the beginning of apoptosis in both MCF7 breast cancer [35] and in T-lymphoblastoid Jurkat cell lines [37]. Namely, the increase in fractal value in breast cancer cells undergoing apoptosis can be measured before any significant change of apoptotic molecular markers are recorded. Loss of fractal geometry could be an early marker of cancer in breast ductal cells [38]. It is arguable that such a transient increase in structure complexity could reflect reorganization and fragmentation processes involving the cytoskeleton as well as the intracellular organelles during apoptosis. 

Our hypothesis was that hyperoxia could induce damage in cancer cells with a high metabolic turnover such as leukemia cells. This damage could change cell shape and induce apoptosis. Fine ultrastructural changes may occur in both the cell and nuclear membranes, as well as in the cytoskeleton or in organelles at the beginning of apoptosis. These changes, previously assessed only by means of qualitative (subjective) parameters, can now be quantitatively measured through a mathematical morphometric approach, based on fractal geometry [39]. 

Several reports have documented the use of fractal analysis for describing irregular morphological traits and changes in the ultrastructural features of membrane organelles, including chromatin and other nuclear components in various types of healthy, pathological, and/or tumor cells [40]. Namely, fractal analysis has helped in discriminating benign from malignant tissues [41], nevi from melanomas [42], and low- from high-grade tumors [43]. Furthermore, fractal studies allow for the discovery of new markers, providing useful insights into cancer identification and prognosis [44,45,46]. Interestingly, transition from tumor to normal phenotype [1], as well as resistance acquisition by cancer cells [47], are mirrored by parallel changes in their fractal dimension. Pathology laboratories have started using fractals in their differential diagnosis [48].

It is likely that early membrane fragmentation processes, associated with precocious apoptosis, will result in a spindle cell-like profile, therefore mimicking an increase in shape complexity. JURKAT cells and CCRF-SB cells both increase their fractal dimensions even in the normoxia group. This could be explained by a continuous change in morphology during the different phases of the cell cycle. 

An increase in fractal dimension suggests a parallel increase in cell morphological complexity, which means that the cell takes up more three-dimensional space, exposing a larger surface, and presenting more receptors to their interacting proteins. On the other hand, heterochromatin’s low fractal dimension would result in a smaller surface area [49].

### 4.3. Microscopic Analysis

Rapid microscopic evaluation shows rough differences between cells under normoxic conditions and those submitted to hyperoxia (Figure 4). We wanted to be able to quantify these cytometric changes. FAB analysis [31], classically used for classifying leukemia of different lineages, was not of any help in our study. We were looking for cytometric changes due to hyperoxia inside the same leukemia cell lines. It seems that the only discriminant criteria to analyze these changes was to identify picnotic nuclei and apoptotic cells [50], which was the case in our study. Insofar as one wishes to analyze apoptosis from a morphological point of view, the pyknotic aspect of the nuclei is one of the most specific criteria. This has been described for a long time and has been recently confirmed. The pyknotic aspect corresponds to a condensation of the chromatin and a contraction of the nucleus.

One of the objectives of our work was to establish a correspondence between specific morphological criteria in optical microscopy and their translation in fractal analysis.

Light and electron microscopy have identified the various morphological changes that occur during apoptosis [51]. During the early process of apoptosis, cell shrinkage and pyknosis are visible by light microscopy [52]. With cell shrinkage, the cells are smaller in size, the cytoplasm is dense, and the organelles are more tightly packed. Pyknosis is the result of chromatin condensation and this is the most characteristic feature of apoptosis.

### 4.4. Caspase-3

Caspase-3 expression was elevated in our cell lines. We have thus demonstrated that hyperoxia by itself can induce apoptosis. Caspase activity was different in the T and B-line cells. T cells seem less resistant to apoptosis as caspase was found to have increased at 6 h whereas in CCRF-SB cells, caspase first decreased (6 h) and then increased (24 h). When we look at Bcl-xl, an anti-apoptotic marker, its activity only increased at 6 and 24 h after hyperoxia in JURKAT cell lines whereas it remained unchanged in CCRF-SB cells. This could be explained by the different time frame of apoptosis between these two cell lines. B cells are produced in the bone marrow. The precursors of T cells are also produced in the bone marrow but leave the bone marrow and mature in the thymus. Oxygen concentrations (the thymus being grossly hypoxic) [53] could be different in these two habitats and could also account for the difference in resistance to hyperoxia.

### 4.5. Oxygen Toxicity

Several discussions on how to evaluate oxygen toxicity in humans are ongoing, and a matter of debate; newly developed approaches have been very recently analyzed, implying new calculations, markers, and also dietetic approaches [54,55,56,57]. In our setting, it is very complicated to set direct links between pulmonary and neurological oxygen toxicity from a clinical point-of-view. Nevertheless, we indirectly relied on actual data showing that the cellular reactions in healthy subjects undergoing a prolonged higher oxygen inspired fraction recover faster (in the first 24 h) and show no permanent damage, as shown from saturation studies (particularly on the decompression phase where the FiO2 is higher for several days) through the erythropoietin rebound reaction [58,59]. Furthermore, intermittent oxygen exposure in laboratory animals has been shown to be beneficial on tumor suppression with or without adjuvant chemotherapy [4,26,60]. The approach of high oxygen levels in septic shock patients has also been discussed and the conclusions are somewhat equivocal without definitive conclusion on the benefit–risk relation of such therapy [61,62]. ROS production is hard to measure directly in cells even though one could use nitrated albumin in whole organs to achieve this. There is, to our best knowledge, no reliable way to measure intracellular antioxidant status. Surrogates are possible, but only partially reflect oxidative stress, and biomarkers have a wide variability [63].

## 5. Conclusions

Our results showed that exposure of T and B cell line cultures to transient normobaric hyperoxia induced apoptosis (elevated caspase-3) as well as significant and precocious modifications in cell complexity, as demonstrated by an increase in fractal dimension and morphological analysis. Such morphological alterations could be due to several molecular mechanisms, with rearrangements in the cancer cell eventually leading to cell cycle inhibition and apoptosis. The exact reasons leading to cell cycle blocking need to be further investigated.

## Figures and Tables

**Figure 1 biomolecules-10-00282-f001:**
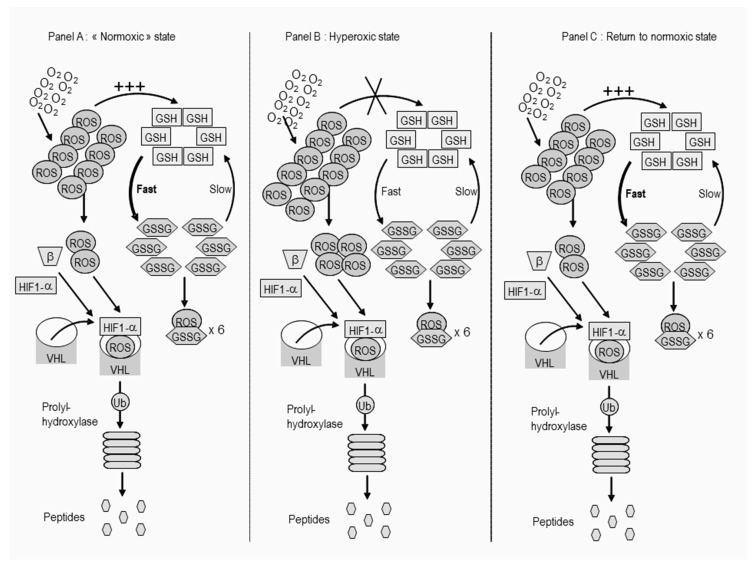
Intracellular mechanisms postulated for the ‘‘Normobaric Oxygen Paradox’’ in the hyperoxic cancer cell. We hypothesize that in the normoxic state, the glutathione synthetase enzyme activity is already at its maximum, thus, an O_2_ increase cannot induce GSH increase and therefore leads to high levels of reactive oxidant species (ROS). Returning to the normoxic state (of a cancer cell) implies that GSSG needs to be reduced back to GSH. This process is rather slow (glutathione reductase works at the expense of NADPH and is thus limited by the conversion rate of glucose). It takes a considerable amount of time before a sufficient level of GSH is attained to restore the optimal GSH-to-GSSG ratio. The inactivation of HIF-1 should be maintained for a longer period of time as a result of this process. (Modified and reproduced with permission from [27].).

**Figure 2 biomolecules-10-00282-f002:**
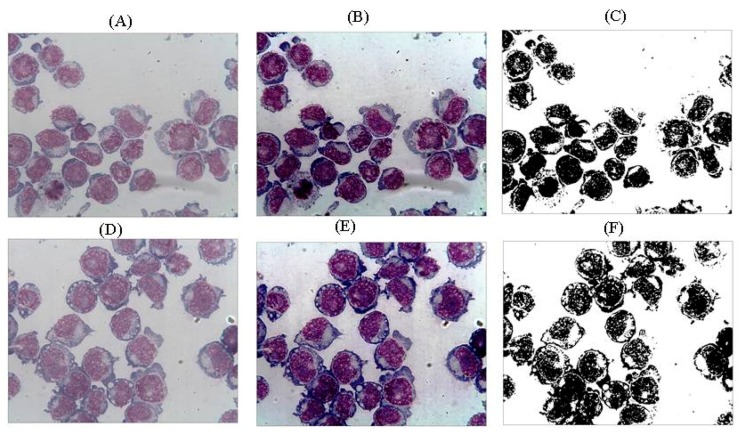
(**A**) Jurkat cells: Normoxia image original version, (**B**) Jurkat cells: Normoxia image after applying enhancing operation. Color histogram is between 0 and 255, (**C**) Black–White Jurkat cells: Normoxia image after applying a threshold at 80. Fractal dimension is calculated for this image by applying the box-counting method with linear increasing box size, and slope analysis is done by using Harfta software. Fractal dimension value after all operations is 1.75 (**D**) Jurkat cells: Hyperoxia cell image original versions, (**E**) Jurkat cells: Hyperoxia image after applying enhancing operation. Color histogram is between 0 and 255, (**F**) Black–White Jurkat cells: Hyperoxia image after applying a threshold at 80. Fractal dimension is calculated for this image applying the same methods. Fractal dimension value after all operations is 1.80.

**Figure 3 biomolecules-10-00282-f003:**
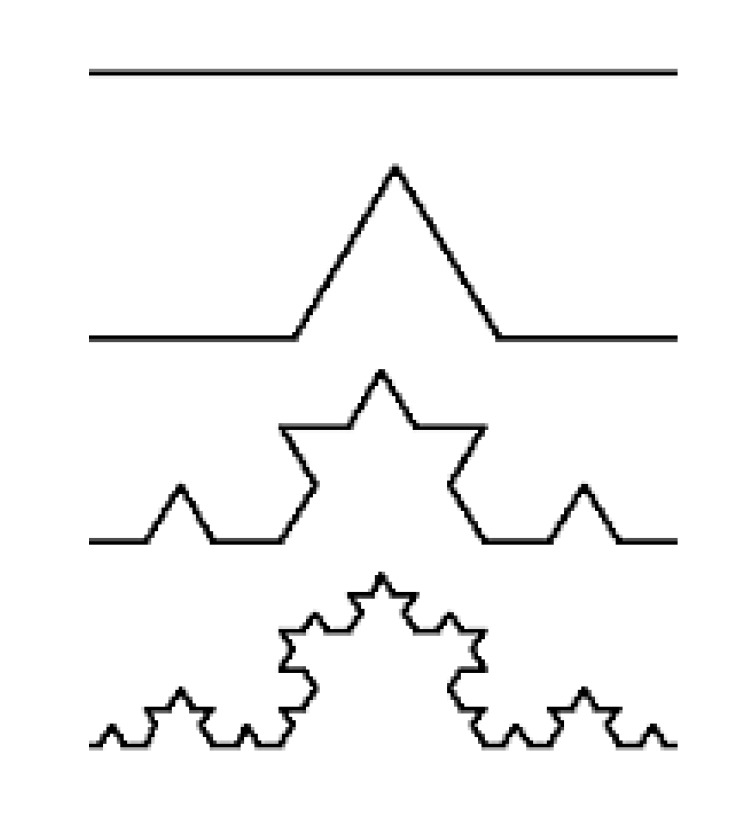
The Koch snowflake.

**Figure 4 biomolecules-10-00282-f004:**
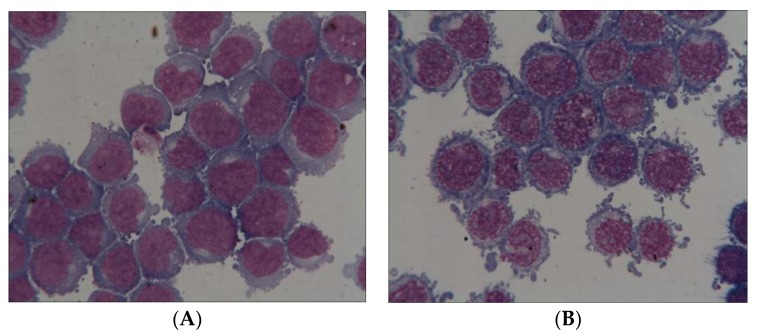
JURKAT cells: Image after normoxia (**A**) and after 18 h of hyperoxia (**B**).

**Figure 5 biomolecules-10-00282-f005:**
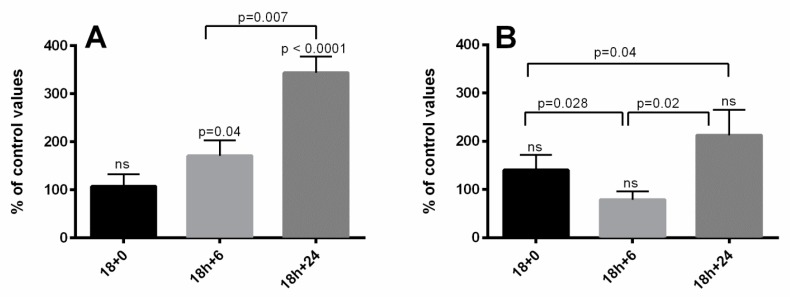
Evolution of Caspase-3 (17-19kDA) as percent variation from baseline in JURKAT (**A**) and CCRF-SB (**B**) cells after an 18-h hyperoxic exposure as compared to normoxia.

**Figure 6 biomolecules-10-00282-f006:**
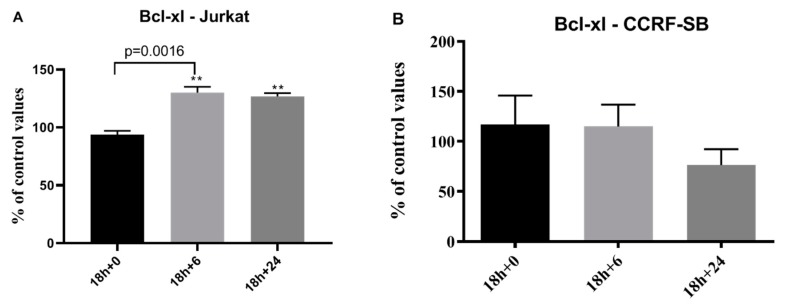
Evolution of Bcl-xl in JURKAT (**A**) and CCRF-SB (**B**) cells after 18 h of hyperoxic exposure as compared to normoxia, **= *p* < 0.01 from the baseline.

**Figure 7 biomolecules-10-00282-f007:**
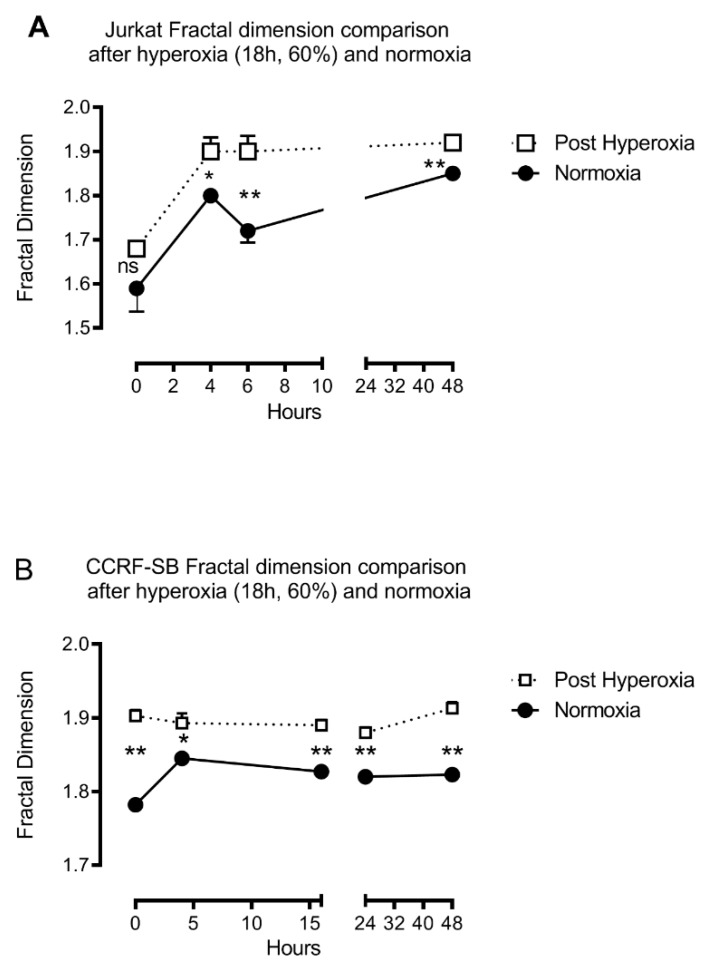
Comparison of the fractal dimensions of JURKAT (**A**) and CCRF-SB (**B**) cells over time after 18 h of hyperoxic exposure as compared to normoxia. Cells were in the same culture medium and were monitored for 48 h, **= *p* < 0.01.

**Figure 8 biomolecules-10-00282-f008:**
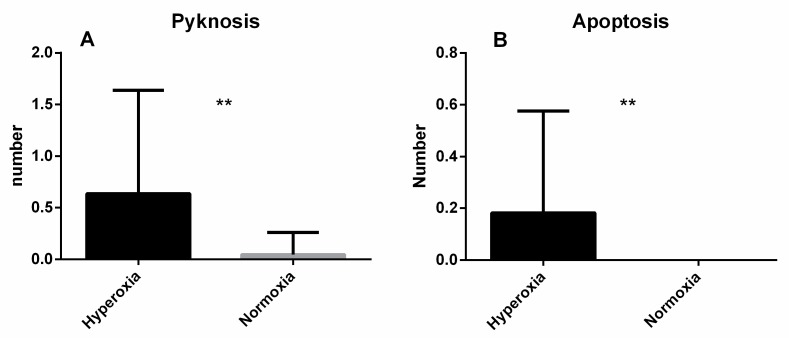
Comparison between the number of pyknotic nuclei (**A**) and apoptotic cells (**B**) in a JURKAT cell line population between the hyperoxia and normoxia groups, as assessed by microscopic analysis (*p* = 0.006). Data are expressed as mean ± SEM.

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
