# Peer review of "Hyperoxia Alters Ultrastructure and Induces Apoptosis in Leukemia Cell Lines"

_biomolecules, 2020, doi:10.3390/biom10020282_

Round 1

Reviewer 1 Report

Please proofread with more clear english construction.(minor)

Author Response

Good Morning,

Thank you for your comments.

English has been externally edited.

All the best

David De Bels

Reviewer 2 Report

The topic of this manuscript falls within the scope of Biomolecules.The purpose of the  study was to investigate the role of transient hyperoxia on the  outcome of non-solid tumor cells such as leukemia line cells.  The Authors showed showed that exposure of T and B cell line cultures to transient normobaric hyperoxia  induced apoptosis  as well as a significant and precocious modifications in cell  complexity. These features are associated to changes in structure (pycnotic nucleus and apoptosis). Such morphological alterations could   leading to cell cycle inhibition and apoptosis. They also showed that T cells seem less resistant to hyperoxia than B cells. The topic of presented study is very important and interesting. The examination the effects of hyperoxia on cancer cells may be useful in the treatment of some tumors. The Authors used appropriate statistic methods. Th discussion is concise but it contains the needed information. The conclusions are consistent with presented evidence and arguments.

the strength of this paper: very interesting and important topic, which gives us new information about influence of hyperoxia on cancer cells. This knowledge may be use in the future in cancer treatment; introduction-relevant and concise; material and methods-the right choice of methodology methods, which was presented in comprehensible way; the obtained results are presented in the form of figures, which  are clear and easy to understand; the discussion- supports the results properly and refers to the current literature in appropriate manner; the conclusions- based on the        obtained results;

weakness of this paper: some small letter errors

Line p. 70

Author Response

(The authors gave the same response as above.)

Reviewer 3 Report

Thank you for having a chance of reviewing the interesting manuscript. This study seems to evaluate the effect of hyper-oxygen status on the apoptosis of leukemia cells. It is interesting. However, there are some aspects that need to be investigated clearly and be readdressed.

The main issues are

The biggest issue seems to be related to oxygen toxicity. Over-production of ROS may be less toxic in single cell observations, but in other organs, it may be very toxic. Did you have any result of the research or evidence regarding the oxygen exposure condition of your research? I think that it would be better if your research could propose the level of ROS production and the change of antioxidant amount. The results of pycnosis are presented as a basis for cell damage and relating apoptosis. Although this is an intuitive analysis, I wonder how pycnosis correlates with actual apoptosis. Therefore, I think that you should add the reason why you use this technique kin the section of “introduction”.

Minor issues are

There are no C and D marks in Figure 5. Figure 6 is two. There are no A and B marks in Figure 7.

Conclusion: It is interesting that hypoxia can alter the leukemia cell lines ultrastructure, but I think we need to add some additional information on whether this is involved in cell apoptosis.

Author Response

Good Morning,

Thank you for your comments.

Please find systemic answers to your comments hereunder.

All the best

David De Bels

Major issues are:

The biggest issue seems to be related to oxygen toxicity. Over-production of ROS may be less toxic in single cell observations, but in other organs, it may be very toxic. Did you have any result of the research or evidence regarding the oxygen exposure condition of your research?

We fully understand the reviewer concern, in fact several discussions on how to evaluate oxygen toxicity in humans nowadays are still ongoing, and a matter of debate, newly developed approaches are only very recently being analyzed, implying new calculations, markers and also dietetic approaches [1-4].

In our setting it is very complicated to set direct links between pulmonary and neurological oxygen toxicity on the clinical point of view. Nevertheless we were indirectly relying on actual data showing that the cellular reactions in healthy subjects undergoing a prolonged higher oxygen inspired fraction recover fast (in the first 24 hours) and show no permanent damage as shown from saturation studies (particularly on the decompression phase were the FiO2 is higher for several days) as shown by the Erythropoietin rebound reaction [5,6].

Furthermore intermittent oxygen exposure in laboratory animals has been shown beneficial

on tumor suppression together with adjuvant chemotherapy or not [7-9].

The approach of high oxygen levels in septic shock patients has also been discussed and the conclusions are somewhat equivocal without definitive conclusion on the benefit-risk relation

of such therapy [10,11].

I think that it would be better if your research could propose the level of ROS production and the change of antioxidant amount.

ROS production is hard to measure directly in cells even though one could use nitrated albumin in whole organs to achieve this. There is, to our best knowledge no reliable way to measure intracellular antioxidant status. Surrogates are possible but only partially reflect oxidative stress and biomarkers have a wide variability [12].

The results of pycnosis are presented as a basis for cell damage and relating apoptosis. Although this is an intuitive analysis, I wonder how pycnosis correlates with actual apoptosis. Therefore, I think that you should add the reason why you use this technique kin the section of “introduction”. 

Insofar as one wishes to analyze apoptosis from a morphological point of view, the pyknotic aspect of the nuclei is one of the most specific criteria. This has been described for a long time and has been confirmed until recently. The pyknotic aspect corresponds to a condensation of the chromatin and a contraction of the nucleus.

One of the objectives of our work was to establish a correspondence between specific morphological criteria in optical microscopy and their translation in fractal analysis.

Light and electron microscopy have identified the various morphological changes that occur during apoptosis [13]. During the early process of apoptosis, cell shrinkage and

pyknosis are visible by light microscopy [14]. With cell shrinkage, the cells are smaller in size, the cytoplasm is dense and the organelles are more tightly packed. Pyknosis is the result of chromatin condensation and this is the most characteristic feature of apoptosis.

Minor issues are:

There are no C and D marks in Figure 5. Figure 6 is two. There are no A and B marks in Figure 7.

This has been corrected.

Conclusion: It is interesting that hyperoxia can alter the leukemia cell lines ultrastructure, but I think we need to add some additional information on whether this is involved in cell apoptosis.

This situation has already been shown in solid tumors [8,9]. According to our results, and given the fact that two different cell lines are responding differently and that Caspase-3 is expressed in both cell lines, we are inclined to think that hyperoxia is indeed involved in cell apoptosis [15].

Ari, C.; Koutnik, A.P.; DeBlasi, J.; Landon, C.; Rogers, C.Q.; Vallas, J.; Bharwani, S.; Puchowicz, M.; Bederman, I.; Diamond, D.M., et al. Delaying latency to hyperbaric oxygen-induced CNS oxygen toxicity seizures by combinations of exogenous ketone supplements. Physiol Rep 2019, 7, e13961, doi:10.14814/phy2.13961. Arieli, R. Calculated risk of pulmonary and central nervous system oxygen toxicity: a toxicity index derived from the power equation. Diving Hyperb Med 2019, 49, 154-160, doi:10.28920/dhm49.3.154-160. Arieli, R. Pulmonary oxygen toxicity in saturation dives with PO2 close to the lower end of the toxic range - A quantitative approach. Respir Physiol Neurobiol 2019, 268, 103243, doi:10.1016/j.resp.2019.05.017. Wingelaar, T.T.; van Ooij, P.A.M.; Brinkman, P.; van Hulst, R.A. Pulmonary Oxygen Toxicity in Navy Divers: A Crossover Study Using Exhaled Breath Analysis After a One-Hour Air or Oxygen Dive at Nine Meters of Sea Water. Front Physiol 2019, 10, 10, doi:10.3389/fphys.2019.00010. Kiboub, F.Z.; Balestra, C.; Loennechen, O.; Eftedal, I. Hemoglobin and Erythropoietin After Commercial Saturation Diving. Front Physiol 2018, 9, 1176, doi:10.3389/fphys.2018.01176. Kiboub, F.Z.; Mollerlokken, A.; Hjelde, A.; Flatberg, A.; Loennechen, O.; Eftedal, I. Blood Gene Expression and Vascular Function Biomarkers in Professional Saturation Diving. Front Physiol 2018, 9, 937, doi:10.3389/fphys.2018.00937. Lee, H.Y.; Kim, I.K.; Lee, H.I.; Lee, H.Y.; Kang, H.S.; Yeo, C.D.; Kang, H.H.; Moon, H.S.; Lee, S.H. Combination of carboplatin and intermittent normobaric hyperoxia synergistically suppresses benzo[a]pyrene-induced lung cancer. Korean J Intern Med 2018, 33, 541-551, doi:10.3904/kjim.2016.334. Raa, A.; Stansberg, C.; Steen, V.M.; Bjerkvig, R.; Reed, R.K.; Stuhr, L.E. Hyperoxia retards growth and induces apoptosis and loss of glands and blood vessels in DMBA-induced rat mammary tumors. BMC Cancer 2007, 7, 23, doi:1471-2407-7-23 [pii]

10.1186/1471-2407-7-23 [doi].

Stuhr, L.E.; Raa, A.; Oyan, A.M.; Kalland, K.H.; Sakariassen, P.O.; Petersen, K.; Bjerkvig, R.; Reed, R.K. Hyperoxia retards growth and induces apoptosis, changes in vascular density and gene expression in transplanted gliomas in nude rats. J Neurooncol 2007, 85, 191-202, doi:10.1007/s11060-007-9407-2. Calzia, E.; Asfar, P.; Hauser, B.; Matejovic, M.; Ballestra, C.; Radermacher, P.; Georgieff, M. Hyperoxia may be beneficial. Crit Care Med 2010, 38, S559-568, doi:10.1097/CCM.0b013e3181f1fe70. Demiselle, J.; Wepler, M.; Hartmann, C.; Radermacher, P.; Schortgen, F.; Meziani, F.; Singer, M.; Seegers, V.; Asfar, P.; investigators, H.S. Hyperoxia toxicity in septic shock patients according to the Sepsis-3 criteria: a post hoc analysis of the HYPER2S trial. Ann Intensive Care 2018, 8, 90, doi:10.1186/s13613-018-0435-1. Katerji, M.; Filippova, M.; Duerksen-Hughes, P. Approaches and Methods to Measure Oxidative Stress in Clinical Samples: Research Applications in the Cancer Field. Oxid Med Cell Longev 2019, 2019, 1279250, doi:10.1155/2019/1279250. Elmer ,S. Apoptosis: A Review of Programmed Cell Death, Toxicol Pathol. 2007 ; 35(4): 495–516. J Eidet JR, Pasovic L, Maria R, Jackson CJ, Utheim TP. Objective assessment of changes in nuclear morphology and cell distribution following induction of apoptosis. Diagnostic Pathology 2014, 9:92  Hacker, G. The morphology of apoptosis. Cell Tissue Res 2000, 301, 5-17, doi:10.1007/s004410000193.

Round 2

Reviewer 3 Report

Thank you for your response.